# Inner centromere localization of the CPC maintains centromere cohesion and allows mitotic checkpoint silencing

Rutger C.C. Hengeveld[1], Martijn J.M. Vromans[1], Mathijs Vleugel[1], Michael A. Hadders[1] & Susanne M.A. Lens[1]

Faithful chromosome segregation during mitosis requires that the kinetochores of all sister chromatids become stably connected to microtubules derived from opposite spindle poles. How stable chromosome bi-orientation is accomplished and coordinated with anaphase onset remains incompletely understood. Here we show that stable chromosome bi-orientation requires inner centromere localization of the non-enzymatic subunits of the chromosomal passenger complex (CPC) to maintain centromeric cohesion. Precise inner centromere localization of the CPC appears less relevant for Aurora B-dependent resolution of erroneous kinetochore–microtubule (KT–MT) attachments and for the stabilization of bi-oriented KT–MT attachments once sister chromatid cohesion is preserved via knock-down of WAPL. However, Aurora B inner centromere localization is essential for mitotic checkpoint silencing to allow spatial separation from its kinetochore substrate KNL1. Our data infer that the CPC is localized at the inner centromere to sustain centromere cohesion on bi-oriented chromosomes and to coordinate mitotic checkpoint silencing with chromosome bi-orientation.

[1] Department of Molecular Cancer Research, Center for Molecular Medicine, University Medical Center Utrecht, Universiteitsweg 100, Utrecht 3584 CG, The Netherlands. Correspondence and requests for materials should be addressed to S.M.A.L. (email: s.m.a.lens@umcutrecht.nl).

Accurate transmission of the genome during cell division requires that the duplicated chromosomes become faithfully segregated in mitosis. Failure to equally segregate the sister chromatids can result in gains and losses of whole chromosomes in the next generation of cells, a condition known as aneuploidy, which is frequently observed in cancer[1]. A prerequisite for error-free segregation is that all sister chromatids bi-orient on the mitotic spindle. This means that specialized multi-protein complexes that assemble at the centromeres of the sister chromatids (that is, kinetochores, KTs) become stably attached to microtubules (MTs) emanating from opposite poles of the mitotic spindle[2]. Chromosome bi-orientation is facilitated by several molecular mechanisms, including centromeric cohesion, the mitotic checkpoint and the chromosomal passenger complex[3–5]. Centromeric cohesin holds the sister-chromatids together until anaphase onset and promotes a back-to-back orientation of sister-kinetochores favoring bipolar microtubule capture[3]. The mitotic checkpoint prevents anaphase onset until all kinetochores have become attached to spindle microtubules and as such provides the necessary time for all chromosomes to bi-orient. And third, the chromosomal passenger complex (CPC), consisting of INCENP, survivin,

borealin and Aurora B kinase, detects and resolves improper kinetochore-spindle connections that, when left uncorrected, would give rise to chromosome mis-segregations[6]. Detachment of incorrect kinetochore–microtubule (KT–MT) connections is a consequence of Aurora B-mediated phosphorylation of outer-kinetochore substrates, including components of the KMN (KNL1, MIS12 and NDC80 complex) network, which directly interact with microtubules[7,8]. Upon chromosome bi-orientation, the opposing pulling forces of the attached KT–MTs are resisted by centromeric cohesin, resulting in tension across sister kinetochores. Tension is thought to pull the outer-KT substrates out of the sphere of influence of Aurora B, resulting in the stabilization of bi-oriented attachments and anaphase onset[9–11]. Central to this 'spatial separation' model is the inner centromere localization of Aurora B. However, there is ongoing debate whether this confined localization of Aurora B is indeed essential for chromosome bi-orientation[6]: apart from its localization at the inner centromere, a small pool of active Aurora B at or near the kinetochore has been described in mammalian cells and suggested to control KT–MT stability[12,13]. Moreover, tension was shown to directly stabilize in vitro reconstituted KT–MT attachments[14], and Aurora B localization

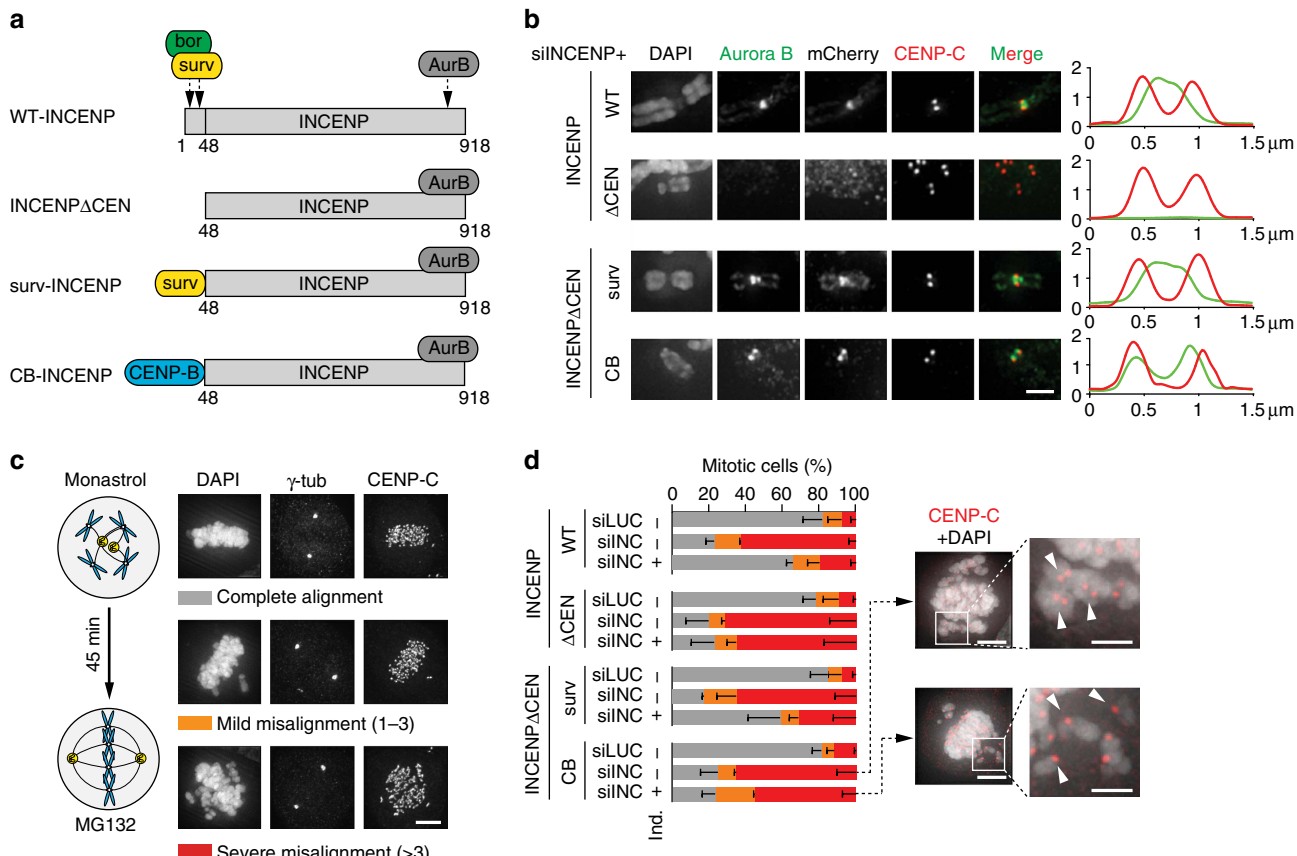

**Figure 1 | Inner centromere localization of the CPC is required for stable chromosome bi-orientation.** (**a**) Scheme of human CPC (bor, borealin; surv, survivin; AurB, Aurora B) and of INCENPΔCEN (deletion of aa 1–48), survivin-INCENPΔCEN (surv-INCENP) and CENP-B-INCENPΔCEN (CB-INCENP). In all experiments HeLa Flp-In T-REx cells expressing the indicated mCherry-tagged INCENP variants were used. + ind., expression induced by doxycycline, − ind., no induction of expression. (**b**) IF of Aurora B, RFP (to detect mCherry), and CENP-C on chromosome spreads of nocodazole treated cells. 1D line graphs of Aurora B (green) and CENP-C (red) are shown on the right. Scale bar, 2 μm. Of note, although surv-INCENP and Aurora B accumulate at the inner centromere, in surv-INCENP-expressing prometaphase cells we also observed some localization of surv-INCENP and Aurora B over the chromosomal arms. Since survivin directly interacts with pH3T3 (refs 47,48) and we frequently detected pH3T3 along the chromosomal arms, this most likely explains the additional arm localization. (**c**) Scheme of the bi-orientation assay (that is, release from a monastrol-induced mitotic arrest into medium containing MG132) and examples of the alignment categories. Scale bar, 5 μm. (**d**) Cells were transfected with siRNAs for Luciferase (siLUC) or INCENP (siINC) and subjected to the bi-orientation assay and chromosome alignment was assessed (n = 2 exp., ≥150 cells per condition, error bars are s.e.m.). Representative images of two conditions (scale bar, 5 μm), and enlargements of selected image regions are shown on the right (scale bar, 2 μm). DNA is visualized using DAPI.

to the inner centromere appeared not to be required for faithful chromosome segregation in budding yeast or for viability of chicken DT40 cells[15,16]. This raised the question what function is executed by the inner centromere pool of Aurora B. Using various INCENP fusion proteins we subtly manipulated the chromosomal localization of Aurora B in human cells and found that, similar to the situation in budding yeast, precise inner centromere localization of the kinase was not required for either error correction or the stabilization of bi-orientated attachments. Yet, we found that, inner centromere localization of Aurora B supported silencing of the mitotic checkpoint once all chromosomes had bi-oriented. In addition, we found that inner centromere localization of the non-enzymatic subunits of the CPC appeared to be essential to maintain centromeric cohesion after chromosome bi-orientation and hence to prevent precocious sister chromatid separation before satisfaction of the mitotic checkpoint.

## Results

**Stable bi-orientation requires inner centromere localization of the CPC.** To study the function of inner centromere-localized Aurora B during mammalian mitosis, we made use of HeLa cell lines that ectopically expressed variants of the CPC scaffold protein INCENP from an inducible promoter, in conjunction with siRNA mediated knockdown of endogenous INCENP (Fig. 1a, Supplementary Fig. 1a). The N-terminal inner centromere-targeting domain (CEN-box, amino acids 1–47) of INCENP, which interacts with the CPC members survivin and borealin[17], was either deleted (INCENPΔCEN) or replaced with different targeting moieties: Survivin (surv-INCENP), which re-localizes Aurora B to the inner centromere, or the centromere-targeting domain of CENP-B (CB-INCENP), which changes the position of Aurora B from predominantly inner centromeric to more prominent near the kinetochore. (Fig. 1a,b)[11,18,19]. Although some Aurora B remained associated with centromeric heterochromatin in CB-INCENP expressing cells, we will refer to this CB-INCENP-shifted pool of Aurora B as 'kinetochore proximal'.

Analysis of INCENPΔCEN expressing cells, confirmed that removal of the CEN-box disrupted the inner centromere localization of Aurora B, similar to what has been reported for Sli15-delta N-terminus (Sli15-dNT), its analogue in S. cerevisiae (Fig. 1b, Supplementary Fig. 1b,c)[16,19,20]. Unlike Sli15-dNT, which supported chromosome bi-orientation in budding yeast, INCENPΔCEN did not rescue chromosome alignment in human cells lacking endogenous INCENP. In fact, we found a strong correlation between the inner centromere localization of Aurora B and stable chromosome bi-orientation in human cells (Fig. 1b–d).

**Loss of inner centromere localized CPC weakens centromeric cohesion.** Close inspection of the misaligned chromosomes in cells with kinetochore-proximal Aurora B (that is, CB-INCENP-expressing cells) revealed the appearance of single sister chromatids. This was in marked contrast to cells lacking INCENP in which the misaligned chromosomes appeared as sister chromatid pairs (Fig. 1d, inset). This suggested that CB-INCENP expressing cells experienced problems in maintaining sister chromatid cohesion after bi-orientation, and we hypothesized that the inner centromere pool of Aurora B might be needed to stabilize centromeric cohesion[21]. Indeed, chromosome spreads of INCENP-depleted cells revealed that sister chromatids behaved as 'railroads' (Fig. 2a,b), a phenotype seen in certain cohesinopathies, and explained by reduced centromeric cohesion, while most likely retaining at least some cohesion

along the chromosomal arms, given the juxtaposition of the sister chromatids[22,23] (Fig. 2a). The railroad chromosome phenotype was rescued after expression of WT-INCENP and surv-INCENP, but these chromosomes were still present in INCENP-ΔCEN expressing cells (Fig. 2b). This observation supports earlier evidence that Aurora B kinase activity is involved in cohesin removal from the chromosomal arms during prophase[24–27], but is also needed for protection of centromeric cohesin[21,26,28]. The observation that INCENP-ΔCEN, which in human cells is mainly cytosolic and on certain fixation conditions slightly visible on chromatin (Supplementary Fig. 5), failed to rescue the cohesion defects induced by INCENP depletion (Fig. 2a,b), suggested that both the loss of arm cohesion, and as well as the cohesion protection at the centromere required robust chromosomal association (and maybe clustering) of INCENP and Aurora B. In cells expressing CB-INCENP we also no longer observed railroad chromosomes, but instead found an increase in the number of mitotic cells with fully separated sister chromatids (<5% for WT-INCENP versus >20% for CB-INCENP expressing cells, Fig. 2a,b). Remarkably, the fraction of cells with fully separated sister chromatids increased when the monastrol treatment, used to synchronize cells in mitosis, was prolonged from 7 to 16 h (~24% for WT-INCENP versus ~61% for CB-INCENP expressing cells, Figs 2c and 4g). This increase may be explained by a slow but gradual degradation of securin during a prolonged mitotic arrest, similar to what has been observed for cyclin B[29,30]. This most likely liberates a number of separase molecules from securin inhibition, which can cleave some of the cohesin rings, thereby 'weakening' sister chromatid cohesion.

Collectively, our experiments suggest that CB-INCENP-localized Aurora B is proficient in resolving chromosomal arm cohesion, but insufficient in maintaining centromere cohesion. In line with these findings, we observed that loss of the CPC from the inner centromere correlated with a loss of the cohesin protector SGO1 from this site in prometaphase cells. In INCENP knockdown or INCENPΔCEN expressing cells, SGO1 was absent from both the inner centromere and kinetochores (Fig. 3a), in agreement with previous work[26,31]. Inner centromere localization of SGO1 was restored by expression of WT-INCENP and surv-INCENP, but not by expression of CB-INCENP (Fig. 3a). In the latter SGO1 was predominantly found at kinetochores (Fig. 3b), a SGO1 pool that does not support centromeric cohesion[32–34]. Thus our data show that inner centromere localized CPC correlates with inner centromere positioning of SGO1 and maintenance of centromeric cohesion after chromosome bi-orientation.

**WAPL depletion rescues bi-orientation in cells with kinetochore-proximal Aurora B.** We then asked whether reduced centromeric cohesion was causing the bi-orientation defect in CB-INCENP-expressing cells (Fig. 1d). To test this, we prevented cohesin removal by knockdown of the cohesin release factor WAPL or by overexpression of a sororin mutant (sororin-9A) that acts as a constitutive WAPL inhibitor (Fig. 4a,b)[35–38]. Indeed, retention of cohesin rescued chromosome alignment in CB-INCENP expressing cells (Fig. 4c,f). In our hands, WAPL depletion was slightly more effective in restoring chromosome bi-orientation than WAPL inhibition via overexpression of sororin-9A (Fig. 4c,f). On the other hand in an INCENP knock-down background, WAPL depletion appeared to exacerbate the bi-orientation defect (Fig. 4c), most likely because retention of arm cohesin delocalizes residual CPC from the inner centromere[39]. Remarkably though, even with Aurora B near kinetochores (Fig. 1b), bipolar, tension-generating, cold-stable end-on KT–MT attachments could be established when

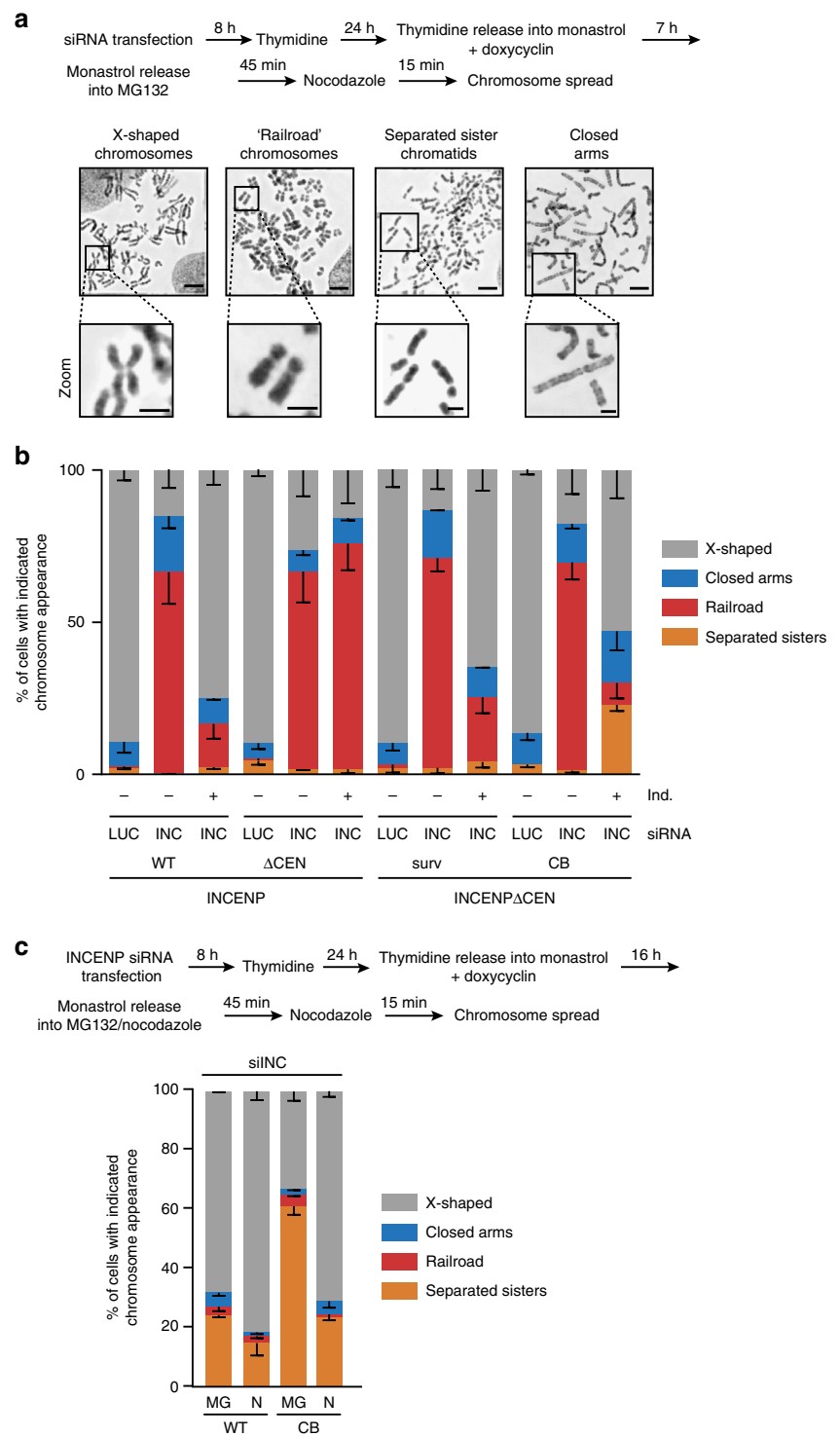

**Figure 2 | Loss of inner centromere localized CPC weakens centromeric cohesion. (a,b)** Cells were transfected with the indicated siRNAs, and synchronized according to the scheme (**a**), After the 45 min. accumulation in MG132 chromosome spreads were prepared (**a**) and the percentage of cells in which all chromosomes appeared as 'X-shaped', 'railroads', with 'closed arms', or in which all sister chromatids were fully separated, was quantified (**b**, $n = 2$ exp., $\geq 185$ cells per condition, error bars are s.e.m.). Representative images of the different categories are shown in **a**. Scale bars, 5 µm, resp. 2 µm in zoom. (**c**) Cells were transfected with siINCENP and treated according to the depicted scheme. After a 16 h monastrol, treatment cells were either released into MG132 (MG) or into nocodazole (N). Chromosome spreads were scored for the indicated chromosome appearance ($n = 2$ exp., $\geq 210$ cells percondition, error bars are s.e.m.).

WAPL was depleted (Fig. 4d, Supplementary Fig. 2a,b). In agreement with this, the N-terminal tail of the Aurora B kinetochore substrate and microtubule binding protein HEC1/NDC80, was also no longer phosphorylated (Fig. 4e)[12]. Since

Aurora B was capable of phosphorylating HEC1 Ser44 in CB-INCENP expressing cells in which microtubules were depolymerized by nocodazole, the absence of HEC1 phosphorylation in metaphase was unlikely to be due to a loss of

function of the fusion protein (Supplementary Fig. 2c). Altogether, this argued that the bi-orientation defect observed after Aurora B displacement in CB-INCENP expressing cells, was a consequence of weakened centromeric cohesion that is unable to resist the opposing pulling forces originating from the attached KT–MT, a phenomenon known as cohesion fatigue[40]. In line with this, precocious sister chromatid separation in CB-INCENP expressing cells was most prominent, when cells were allowed to bi-orient on the mitotic spindle (release from monastrol into MG132) than when microtubules were absent (release from monastrol into nocodazole)(Fig. 2c).

**Stable centromeric cohesion requires the CEN-box of INCENP.** We next considered two possibilities why centromeric cohesion was less robust in CB-INCENP expressing cells: stable centromeric cohesion requires a pool of active Aurora B at the inner centromere, or it requires presence of the N-terminal CEN-box of INCENP, missing in CB-INCENP. The latter possibility predicts that expression of the CEN-box would be sufficient to rescue chromosome alignment in CB-INCENP expressing cells. In fact, this prediction appeared to be true; re-introduction of the CEN-box in CB-INCENP expressing cells rescued chromosome alignment to a similar extent as expression of a sororin-9A mutant,

which has been reported to bypass the requirement for SGO1-PP2A in maintaining centromeric cohesion (Fig. 4f, Supplementary Fig. 3a–d)[25,37]. Fusing an active form of Aurora B (Baronase[41]) onto the CEN-box, guided Baronase localization to the inner centromere but did not further improve chromosome bi-orientation, indicating that the CEN-box itself is critical in stabilizing centromeric cohesion (Fig. 4f, Supplementary Fig. 3b). Indeed, analysis of chromosome spreads revealed that either transduction of sororin-9A or of the CEN-box in CB-INCENP expressing cells diminished the number of cells with fully separated sister chromatids and increased the fraction of cells with X-shaped chromosomes (Fig. 4g). Of note, the observation that sororin-9A expression predominantly rescued centromeric cohesion in CB-INCENP expressing cells instead of both arm and centromere cohesion as observed on WAPL depletion (Fig. 4b,g), is similar to what has been reported for the level of cohesion rescue by non-phosphorylatable sororin mutants in SGO1 depleted cells[25,37]. To test whether stabilization of centromeric cohesion involved inner centromere localized SGO1. We co-depleted endogenous INCENP and SGO1 in CB-INCENP expressing cells and re-introduced exogenous SGO1 or a fusion protein consisting of the CEN-box and SGO1 (CEN-SGO1)(Fig. 4h, Supplementary Fig. 3b). While exogenous SGO1 was capable of restoring chromosome bi-orientation in SGO1-

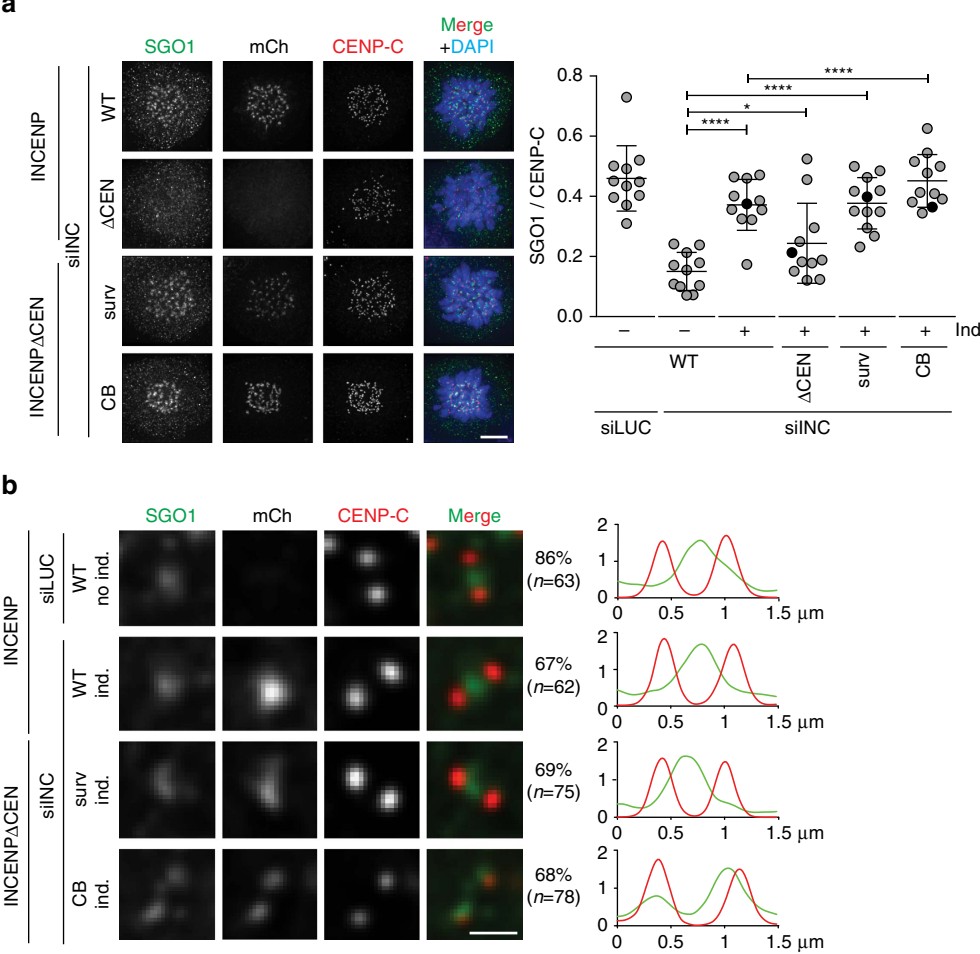

**Figure 3 | SGO1 is predominantly localized at kinetochores in CB-INCENP expressing cells.** (**a**) IF and quantification of fluorescence intensities (FI) of SGO1/CENP-C on centromeres and/or kinetochores in the depicted cell lines, transfected with the indicated siRNAs, and blocked in mitosis using STLC (1 exp. out of 2, 11 cells per condition, bars: mean ± s.d., ns, not significant; *P < 0.05; ****P < 0.0001; unpaired t test). DNA is visualized using DAPI. Scale bar, 5 μm. (**b**) Representative sister kinetochores with SGO1, mCherry and CENP-C and 1D line graphs of SGO1 (green) and CENP-C (red) of cells transfected with the indicated siRNAs, blocked in mitosis using STLC and expressing INCENP variants that rescued SGO1 centromere/kinetochore localization (**a**). The percentage of kinetochore pairs showing that particular SGO1 staining pattern is shown on the right side of the images. Scale bar, 1 μm.

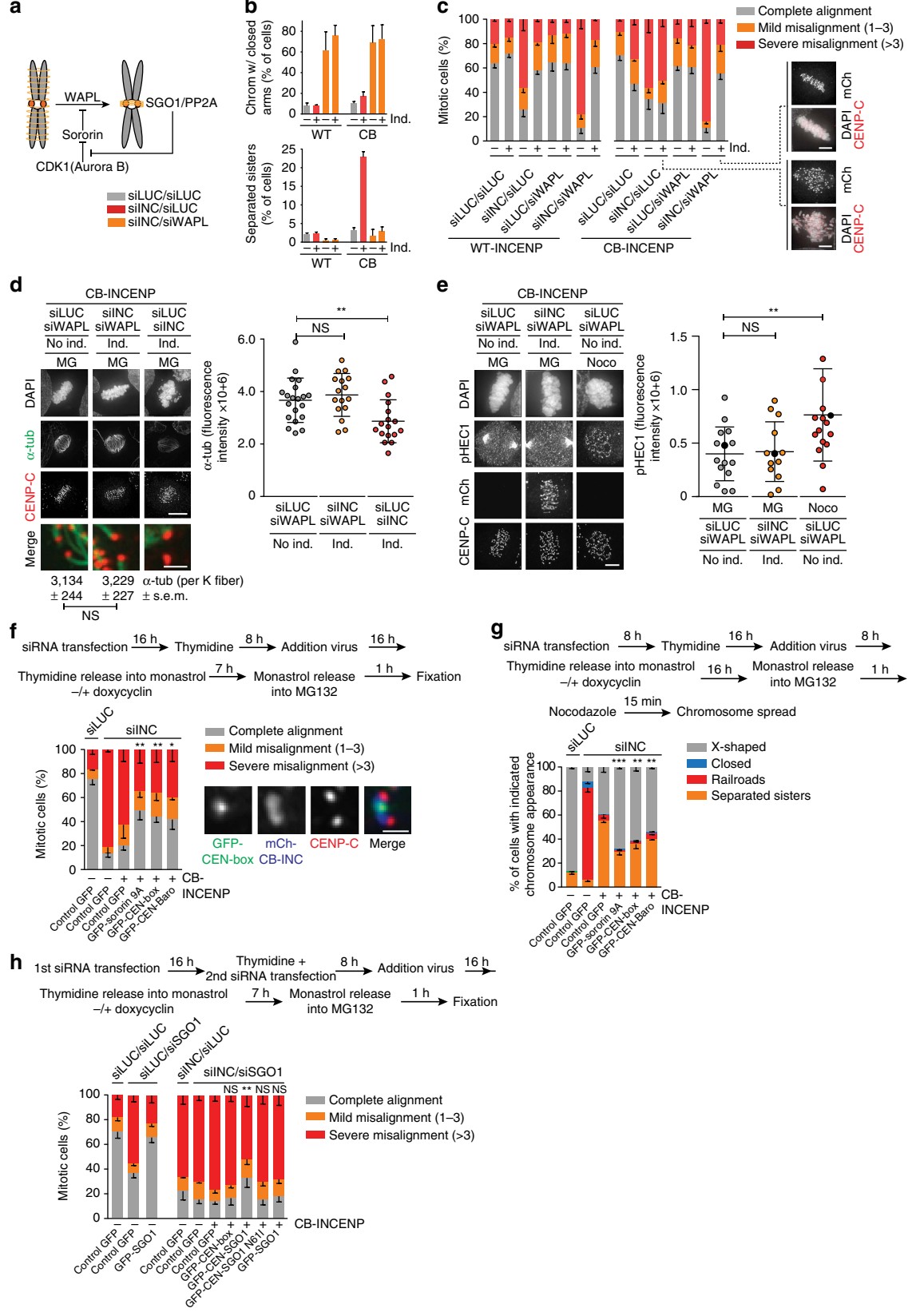

depleted cells, it was incapable in INCENP and SGO1 double knockdown cells (Fig. 4h). Similarly, expression of the CEN-box did not support chromosome alignment in CB-INCENP expressing cells in which both INCENP and SGO1 were depleted (Fig. 4h). However, expression of a CEN-SGO1 fusion protein that localized to the inner centromere when CB-INCENP was expressed (Supplementary Fig. 3b), improved chromosome bi-orientation in INCENP/SGO1 double knockdown cells (Fig. 4h). Importantly, a CEN-SGO1 N61I mutant that does not bind PP2A did not improve chromosome bi-orientation, highlighting the importance of PP2A in SGO1-dependent cohesion protection (Fig. 4a,h)[42–44]. Taken together, our data suggest that the CEN-box in INCENP, either directly or through association with borealin and survivin, is needed to prevent precocious sister chromatid separation on chromosome bi-orientation. Our data further imply that Aurora B itself does not need to reside at the inner centromere to maintain centromeric cohesion or to allow the stabilization of bi-oriented KT–MT attachments (Fig. 4d,g, Supplementary Fig. 2a). We found that Aurora B localized near the kinetochore in CB-INCENP expressing cells, can correct erroneous KT–MT attachments (Fig. 4c), and provide the necessary positive feedback for Haspin-induced H3-T3 phosphorylation, and BUB1-mediated H2A-T120 phosphorylation, both of which are required for inner centromere recruitment of the CEN-box (Supplementary Fig. 3b–d)[45–48]. This raised the question why Aurora B kinase needs to be confined to the inner centromere.

**Inner centromere localization of Aurora B supports mitotic checkpoint silencing.** Microtubules that are stably attached to kinetochores are expected to silence the mitotic checkpoint[49,50]. However, we measured a significant metaphase delay in the CB-INCENP expressing cells in which we had knocked down WAPL (265 min, s.d. = 128 min versus 52 min, s.d. = 27 min and 50 min, s.d. = 29 min, in respectively WAPL-depleted cells with endogenous INCENP or with ectopic WT-INCENP expression, Fig. 5a, Supplementary Fig. 4a). This suggested that despite stable bi-oriented attachments (Fig. 4d, Supplementary Fig. 2a), silencing of the mitotic checkpoint was impaired.

The kinetochore protein KNL1 is an important signalling hub for the mitotic checkpoint. It recruits several checkpoint proteins including BUB1, BUB3, BUBR1 and MAD1, after phosphorylation of its MELT motifs by MPS1 (ref. 51).

Phosphorylation of the KNL1 MELT motifs is antagonized by protein phosphatase 1 gamma (PP1γ), which is recruited to the N-terminal RVSF motif in KNL1 (refs 52,53). Phosphorylation of this RVSF motif by Aurora B hampers PP1γ binding to KNL1 allowing optimal MELT phosphorylation and mitotic checkpoint activity[52,53]. Conversely, RVSF phosphorylation needs to go down in metaphase to allow kinetochore recruitment of PP1γ and silencing of the mitotic checkpoint[53]. We found that the phosphorylation status of the RVSF motif in KNL1 was enhanced on metaphase chromosomes of WAPL-depleted cells with Aurora B localized near kinetochores (Fig. 5b, Supplementary Fig. 2d). This correlated with enhanced MELT phosphorylation, increased kinetochore recruitment of BUB1, and low levels of MAD1 on all metaphase kinetochores (Fig. 5c–e, Supplementary Fig. 4e). Altogether, our data suggest that inner centromere localization of Aurora B is needed to spatially separate the kinase from its substrate KNL1, to promote mitotic checkpoint silencing on chromosome bi-orientation. As mentioned, in contrast to KNL1, the N-terminal tail of HEC1 was no longer phosphorylated and KT–MT attachments were stable in CB-INCENP expressing, WAPL-depleted cells (Fig. 4d,e, Supplementary Fig. 2a). This either implies that spatial separation of Aurora B from its KT substrates may be a more relevant mechanism for mitotic checkpoint silencing than to allow the dephosphorylation of the HEC1 N-terminus and the stabilization of bi-oriented attachments, or that due to increased tension (Supplementary Fig. 2b), even CB-INCENP positioned Aurora B can no longer reach HEC1. To discriminate between these two possibilities we expressed a fusion protein of INCENPΔCEN and the kinetochore protein MIS12 (MIS12-INCENP), to force localization of Aurora B at kinetochores[11]. However, in an INCENP knock-down background, the MIS12-INCENP expression levels did not restore phosphorylation of CENP-A and DSN1 to levels comparable to WT-INCENP expressing cells (Supplementary Fig. 5a–c), making the knock-down, add-back set-up unsuitable. We chose to perform experiments in the presence of endogenous INCENP and in the absence of WAPL, since this would allow us to analyse the mere consequence of maintaining a small pool of kinetochore-localized Aurora B while avoiding KT–MT attachment problems due to overall reduced levels of active Aurora B, or due to cohesion loss (Supplementary Figs 5c,6a). Cells were synchronized in G2 and subsequently

**Figure 4 | Cohesin maintenance rescues chromosome bi-orientation and promotes stable KT–MT attachments in CB-INCENP expressing cells.** (**a**) Scheme of the regulation of WAPL-dependent cohesin removal from chromosomal arms. (**b**) Quantification of the % of cells with chromosomes with fully closed arms (upper graph) and with fully separated sisters (lower graph) as a measure for efficiency of the siRNA-mediated WAPL knockdown (n = 2 exp., ≥201 cells per condition, error bars are s.e.m.). Note that the conditions without siWAPL are also presented in Fig. 2b. (**c**) Cells were transfected with the indicated siRNAs and subjected to the bi-orientation assay (Fig. 1c). Chromosome alignment was assessed (n = 3 experiments, ≥159 cells per condition, error bar is s.e.m., ns = not significant; ****P < 0.0001; Chi-squared test for comparison of the indicated groups for % complete alignment). Representative images of two conditions are shown. (**d**) IF for α-tubulin and CENP-C in cells transfected with the indicated siRNAs and subjected to the bi-orientation assay followed by ice-cold treatment (Scale bar, 5 μm). FI quantifications of spindle α-tubulin (1 out of 2 exp., ≥16 cells per condition, bars: mean ± s.d., ns = not significant; **P < 0.01; unpaired t test). Numbers below merged images (scale bar: 1 μm) depict the mean FI of ≥17 individual K fibers. (**e**) IF for phospho-HEC1 (Ser44, pHEC1), mCherry and CENP-C (1 out of 2 exp., ≥13 cells per condition, bars: mean ± s.d., ns, not significant; **P < 0.01; unpaired t test). The corresponding data points of the images (scale bar: 5 μm) are coloured black in the graph. DNA is visualized using DAPI. Spindle poles were excluded from the quantification. (**f,h**) Cells ± induction of CB-INCENP, were subjected to the depicted experimental set-up (siSGO1 was co-transfected directly after thymidine addition (**h**)), and chromosome alignment was assessed (n = 3 exp, ≥240 cells per condition in (**f**) and ≥208 cells per condition in **h**, error bars are s.e.m., ns = not significant; ***P < 0.001; Chi-squared test for differences between the indicated groups and the control GFP, + CB-INCENP group, for % complete alignment). Localization of GFP-CEN-box is shown in the IF images **f**. Scale bar, 1 μm. (**g**) Cells ± induction of CB-INCENP, were subjected to the depicted experimental set-up and chromosome spreads were scored for the indicated categories (n = 2 exp., ≥312 cells per condition, error bars are s.e.m., **P < 0.01, ***P < 0.001; Chi-squared test for differences between the indicated groups and the control GFP, + CB-INCENP group, for % separated sisters). Note that due to the fixation procedure used for the preparation of chromosome spreads, fluorescence was quenched and we could therefore not select for transduced cells. Hence we determined the frequency of cells with the indicated chromosome appearance in the total cell population. The observed rescue effect observed for sororin − 9A, CEN-box and CEN-Baronase is most likely an underestimation.

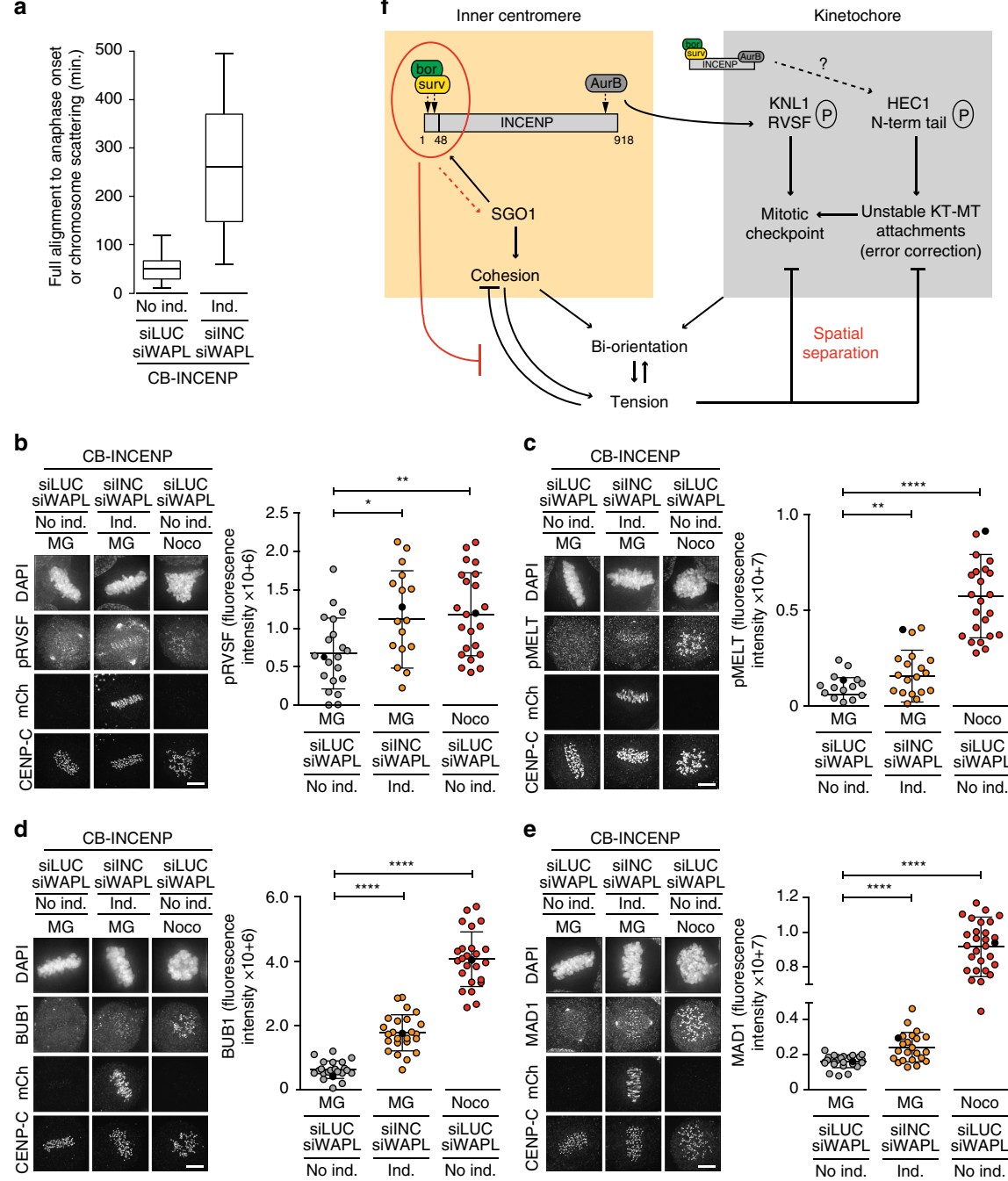

**Figure 5 | Spatial separation of Aurora B from KNL1 coordinates mitotic checkpoint silencing with bi-orientation.** (**a**) Time in metaphase for WAPL- and/or INCENP depleted cells ± CB-INCENP ($n = \geq 21$ cells, see also Supplementary Fig. 4a–d). (**b–e**) IF of mCherry, CENP-C and KNL1-pRVSF-S60 (pRVSF) (**b**), KNL1-pMELT-T601 (pMELT) (**c**), BUB1 (**d**) and MAD1 (**e**) in cells ± induction of CB-INCENP, transfected with indicated siRNAs and subjected to the bi-orientation assay. Quantifications of fluorescence intensities are shown on the right side of each panel (1 out of 2 exp., $n = \geq 18$ cells per condition, bars: mean ± s.d., *$P < 0.05$; **$P < 0.01$; ****$P < 0.0001$; unpaired $t$ test, spindle poles were excluded from the quantifications). The corresponding data points of the images (scale bars, 5 μm) are coloured black in the graphs. (**f**) Model for how the CPC regulates the inner centromere and kinetochore to allow the build-up of tension upon bi-orientation, to counteract cohesion fatigue upon bi-orientation and tension, and to coordinate mitotic checkpoint silencing with chromosome bi-orientation.

released from a CDK1-inhibitor block to allow progression through mitosis (Supplementary Fig. 6b). Unlike WT-INCENP expressing cells, MIS12-INCENP expression resulted in an increased frequency of cells in metaphase. This metaphase accumulation was even more pronounced in MIS12-INCENP expressing cells than in CB-INCENP expressing cells, in which all endogenous Aurora B was redistributed in closer proximity of the KT (Supplementary Fig. 6a,c). This indicated that the

small pool of KT-localized Aurora B in MIS12-INCENP expressing cells was highly effective in preventing mitotic checkpoint silencing and in delaying anaphase onset. Interestingly, when blocking cells in metaphase with MG132 for 90 min., we found that MIS12-INCENP expressing, WAPL depleted cells, remained fully aligned, similar to WT-INCENP and CB-INCENP expressing cells (Supplementary Fig. 6d). This implied that kinetochore-localized Aurora B did not detach

bi-oriented attachments when centromere cohesion was preserved.

## Discussion

Our work demonstrates that in human mitotic cells, inner centromere localization of the CPC is a prerequisite for stable chromosome bi-orientation because it ensures stable centromeric cohesion of the sister chromatids. Stable centromeric cohesion not only promotes tension across sister kinetochores on bi-orientation, but also prevents precocious sister chromatid separation due to cohesion fatigue when bipolar KT–MT attachments have been established (Fig. 5f). Prevention of cohesion fatigue is most likely critical for the first sister-chromatids pairs that have obtained tension-generating, bipolar attachments and that need to 'wait' for the other sister chromatid pairs to become properly connected to the mitotic spindle. The centromere cohesion protective function of the CPC is dependent on the CEN-box of INCENP and requires Aurora B kinase activity to provide the necessary feedback for H3-T3 and H2A-T120 phosphorylation that promotes inner centromere docking of the CEN-box[54–56]. How the CEN-box confers centromeric cohesion stability remains to be resolved. Because direct fusion of survivin to INCENPΔCEN also rescued chromosome cohesion defects and chromosome bi-orientation, we consider it likely that the non-enzymatic CPC core complex[17], consisting of the INCENP CEN-box, survivin and borealin, mediates this cohesion protective function. Moreover, the observation that the localization of the cohesin protector SGO1 was shifted from the inner centromere to kinetochores in CB-INCENP expressing cells, suggests a role for the CPC core complex in either driving SGO1 towards or retaining the cohesin protector at the inner centromere. However, since we were unable to assess the exact SGO1 location in CEN-box reconstituted cells, we cannot exclude the possibility that the CEN-box may also act independent of SGO1 in stabilizing centromeric cohesion, for instance by counteracting the decatenation of centromeric DNA. Interestingly, in budding yeast, the Sli15-dNT (the budding yeast analogue of INCENP-ΔCEN) mutant is synthetic lethal with mcm21Δ, ctf19Δ, and ctf18Δ (ref. 16). Although the mechanistic underpinning of this synthetic lethality is unclear, the observation that proteins encoded by these genes are involved in either recruitment or loading of (pericentromeric) cohesin[57–59], suggested a potential functional connection of chromatin-localized CPC with cohesion[16]. Here we provide experimental evidence that the N-terminus of INCENP, most likely in conjunction with survivin and borealin, is critical for maintenance of centromeric cohesion after chromosome bi-orientation in human cells (Fig. 5f).

Remarkably, Aurora B does not need to be localized at the inner centromere to provide feedback for histone phosphorylation. Neither does Aurora B have to be at the inner centromere for its KT–MT error correction activity or to allow stabilization of correctly attached KT–MTs. This latter finding is in line with the situation in budding yeast in which truncation of the Sli15 N-terminus (which results in localization of the Aurora B homologue Ipl1 to kinetochores and maybe spindle microtubules instead of the inner centromere) only has a very minor effect on the fidelity of chromosome bi-orientation and segregation[16]. However, unlike Ipl1 in budding yeast, inner centromere confinement of Aurora B in human cells appears important to silence the mitotic checkpoint. Failure to silence the checkpoint is lethal in budding yeast[60], however Sli15-dNT mutants are viable[16], strongly suggesting that mitotic checkpoint silencing is not impaired in these mutants. We propose that the inner centromere localization of Aurora B allows it to become spatially separated from its kinetochore substrate KNL1 thereby supporting the recruitment of PP1 (Fig. 5f). We favour this model because both in prometaphase and metaphase Aurora B is mainly detected at the inner centromere. However, since the existence of a small but highly active KT pool of Aurora B has been suggested[12,13], an alternative scenario could be that a KT pool of endogenous Aurora B is responsible for the phosphorylation of KNL1-RVSF. Since this putative KT pool of Aurora B is absent from bi-orientated chromosomes[12], tension-dependent removal of this pool from kinetochores might be a way to allow the recruitment of PP1 by KNL1 and to support mitotic checkpoint silencing (Fig. 5f). Obviously, in CB-INCENP expressing cells the removal of this potential KT pool of Aurora B is bypassed. Although we were unable to visualize PP1 and could therefore not directly assess whether PP1 was absent from metaphase kinetochores in CB-INCENP expressing cells, the observed low levels of MELT phosphorylation, and BUB1 and MAD1 kinetochore levels, strongly suggest PP1 recruitment was hampered. While mitotic checkpoint silencing in budding yeast also requires PP1 kinetochore recruitment by KNL1 (known as Spc105 in budding yeast), there is currently no evidence supporting a role for Ipl1 in counteracting this Spc105-dependent PP1 recruitment[60,61], and this most likely explains why Sli15-dNT mutants can tolerate kinetochore-localized Ipl1.

Finally, the observation that in CB-INCENP expressing cells, bi-oriented attachments were stabilized and HEC1-Ser44 was dephosphorylated, while the RVSF motif in KNL1 was still phosphorylated and thus most likely not recruiting PP1, implies that other phosphatases, or other PP1 regulatory subunits, are involved in the dephosphorylation of the HEC1 N-terminus and in the stabilization of attachments. Moreover, both in CB-INCENP and MIS12-INCENP expressing cells, in which Aurora B was redistributed respectively near or at the kinetochore, chromosome bi-orientation was maintained when cohesin removal was prevented. This re-raises the longstanding question how tension stabilizes KT–MT attachments, since it suggests that spatial separation of the HEC1 N-terminus from Aurora B kinase may not be the main mechanism. Formally, we cannot rule out that, despite being sufficient to prevent anaphase onset, the KT pool of Aurora B maintained in MIS12-INCENP expressing cells was too small to sustain phosphorylation of the more distant HEC1 N-terminus upon bi-orientation. Still, we argue that alternative models explaining how tension stabilizes attachments and how Aurora B discriminates between correct and incorrect KT–MT attachments need to be considered. Such models include direct mechanical stabilization of KT–MT attachments[14], in combination with either a microtubule-associated HEC1 phosphatase or tension-dependent exposure of a HEC1 phosphatase-binding site on the kinetochore.

## Methods

**Cell lines and cell culture.** HeLa cells (ATCC CCL-2) were cultured in Dulbecco's modified Eagle's medium (DMEM, Sigma Aldrich) supplemented with 6% Tetra-cycline Screened HyClone Fetal Bovine Serum (GE Healthcare), 1 mM ultra-glutamine (Lonza) and streptomycin/penicillin (Sigma Aldrich). HeLa Flp-In T-REx cells (gift from S. Taylor, University of Manchester, Manchester, UK) were cultured in the same medium as HeLa cells that included 4 µg ml⁻¹ Blasticidin S (Invitrogen). To generate stable cell lines, the pcDNA5/FRT/TO plasmids encoding VSV-INCENP-mCherry, VSV-INCENPΔCEN-mCherry, VSV-survivin-INCENPΔCEN-mCherry, VSV-CB-INCENPΔCEN-mCherry, VSV-MIS12-INCENPΔCEN-mCherry (verified using sequencing by Macrogen) were co-transfected with pOG44 (Invitrogen) using the standard FuGENE 6 (Promega) transfection protocol. After transfection, cells were selected in medium supplemented with 200 µg ml⁻¹ Hygromycin B (Roche). To generate HeLa VSV-INCENP-mCherry and VSV-CB-INCENPΔCEN-mCherry cells stably expressing H2B-GFP, lentivirus was produced. HEK293T (ATCC, CRL-3216) cells were co-transfected with pWPT-H2B-GFP, pRSV, pMD2-G and pMDLG-I using the X-tremeGENE (Roche) transfection protocol. The transfected HEK293T cells were cultured as described above without Blasticidin S. After 48 h, viruses were

harvested and the VSV-INCENP-mCherry and VSV-CB-INCENPΔCEN-mCherry expressing HeLa Flp-In T-REx cells were used as donor cells for viral transduction. HeLa cell lines expressing LAP-KNL1Δ87-1832 from a doxycycline-inducible promoter at a single integration site have been described and were used to characterize the pMELT-T601 antibody[62]. Spodoptera frugiperda (Sf) 9 insect cells (ATCC, CRL-1711) were cultured at 27 °C in Insect-XPRESS media (Lonza, Basel, Switzerland) supplemented with 5% FBS and penicillin/streptomycin. All cell lines were tested negative for mycoplasma.

**siRNA transfections and cell synchronization.** The following siRNAs were used: siLUC (Luciferase GL2 duplex; Dharmacon/D-001100- 01-20), siINCENP (Dharmacon/3′-UTR: GGCUUGGCCAGGUGUAUAU), siSGOL1 (Dharmacon/J-015475-12: GAUGACAGCUCCAGAAAUU), siWAPL (Dharmacon/J-026287-10: GAGAGAUGUUUACGAGUUU) and siKNL1 (Thermo Fisher Scientific/J-015673–05: GCAUGUAUCUCUUAAGGAA). siRNAs were reverse transfected using HiPerFect (Qiagen) at 20 nM for siLUC, siINCENP, siSGOL1, siKNL1 and 40 nM for siWAPL. Cells were seeded on 12 mm High Precision coverslips (Superior-Marienfeld GmbH & Co) in 24 wells plates. After 16 h of siRNA transfection, cells were synchronized in G1/S-phase by addition of 2.5 μM thymidine (Sigma Aldrich). When SGO1 was co-depleted with INCENP, siINCENP was first reverse transfected and cells were subsequently forward transfected with siSGO1 directly after thymidine addition. After 24 h, cells were released from the thymidine block into medium containing 20 μM S-trityl-L-cysteine (STLC, Tocris), or 100 μM monastrol (Sigma Aldrich). At the same time 1 μg ml⁻¹ doxycycline (Sigma Aldrich) was added to induce protein expression. To assess the capacity to bi-orient chromosomes, monastrol was washed out 7 h later and medium containing either 10 μM MG132, or 0.83 μM nocodazole (Sigma Aldrich) (Calbiochem) was added for 45 min. Alternatively, cells were synchronized in G2 after the thymidine release via treatment with the CDK1 inhibitor RO3306 (5 μM, Calbiochem) for 6 h. Cells are subsequently released from the block by 3× washing with PBS and fixed 2 h later. Reversine (250 nM, Sigma Aldrich) was used to inhibit MPS1.

**Antibodies and immunofluorescence.** For immunofluorescence (IF) of γ-tubulin, cells were fixed for 5 min with 4% PFA (Sigma Aldrich), washed once with PBS and permeabilized with ice-cold methanol. After blocking in PBS containing 0.05% Tween 20 and 3% BSA, coverslips were incubated at room temperature with the following primary antibodies: mouse anti-γ-tubulin (1:500, Sigma Aldrich—T5192), rat anti-RFP (1:500, ChromoTek—5F8, to detect mCherry) and guinea pig anti-CENP-C (1:500, MBL—PD-030, to visualize the kinetochores). For IF of GFP, Aurora B, SGO1, pDSN1, KNL1-pRVSF-KNL1, KNL1-pMELT, pCENPA, MAD1, BUB1, CREST, pH3T3 and pH2A-T120, a brief pre-extraction with 100 mM Pipes pH 6.8, 10 mM EGTA pH 8, 1 mM MgCl₂, 0.2% Triton X-100 (PEM/T) was performed followed by addition of an equal volume of 4% PFA. After 5 min, the mixture was removed and 4% PFA was added for 5 min. After washing once with PBS, the coverslips were blocked as described above and subsequently incubated with rabbit anti-GFP (1:500, custom made), mouse anti-Aurora B (1:1,000, BD Transduction Laboratories–611083) or mouse anti-SGOL1 (1:1,000, Abnova–h00151648), rabbit anti-pCENPA-S7 (1:500, Millipore/Upstate–07-232), rabbit anti-pDSN1-S109 (1:2,000) and rabbit anti-KNL1-pRVSF-S60 (1:1,000, kind gifts of Iain Cheeseman[63]), rabbit anti-KNL1-pMELT-T601 (1:2,000, see below), mouse anti-MAD1 (1:1,000, Merck Millipore–MADE867), rabbit anti-BUB1 (1:1,000, Abcam–AB9000), human CREST (1:2,000, Cortex Biochem–cs-1,058), rabbit anti-pH3T3 (1:2,000, Upstate–07-424) and rabbit anti-pH2A-T120 (1:2,000, Active Motif–39,391). Anti-MAD1 was incubated overnight at 4 °C, whereas all other antibodies were incubated at room temperature for 2 h. Affinity-purified phospho-specific antibody recognizing KNL1-pMELT-T601 was generated by injection of rabbit with KLH-coupled MDLpTESHTSNLGSQC peptide and affinity purification of bleed-out serum (GenScript). For the cold-stable microtubule assay, cells were incubated with ice-cold medium for 5 min. Then the cells were pre-extracted with PEM/T as described above and mouse anti-α-tubulin (1:10,000, Sigma Aldrich–T5168) was used as primary antibody. Goat anti-mouse (A11029) or anti-rabbit (A11034) IgG-Alexa 488, goat anti-mouse (A11031), anti-rabbit (A11036) or anti-rat (A11077) IgG-Alexa 568 (Invitrogen), goat anti-guinea pig (A21450) IgG-Alexa 647 (Invitrogen) and goat anti-human (A21445) IgG-Alexa 647 (Invitrogen) were used as secondary antibodies (all 1:500). DNA was stained with 1 μg ml⁻¹ 4′,6-diamidino-2-phenylindole (DAPI, Sigma) for 2 min. The coverslips were washed once with PBS, dipped in 100% ethanol, dried and mounted onto glass slides using ProLong Antifade Gold (ThermoFisher) mounting media. Images were acquired on a deconvolution system (DeltaVision RT; Applied Precision) with a CoolSNAP HQ2 camera and a × 100/1.40 NA U Plan Apochromat objective (Olympus) using softWoRx software (Applied Precision). All images are maximum intensity projections of deconvolved stacks. For quantifications of immunostainings, all images of similarly stained experiments were acquired with identical illumination settings and analysed using ImageJ (National Institute of Health). An image J macro was used that automatically selected kinetochores, this selection were enlarged with 3 pixels (px), and this region of interest (ROI) was then used to measure fluorescence intensities in different channels. For background subtraction, a selected area surrounding the DAPI signal was selected, this area was enlarged with 4 px (ROI-A) and with 6 px (ROI-B). ROI-A was subtracted from

ROI-B, and this selected region was used as background ROI. For quantification of α-tubulin fluorescence intensity per individual K-fibre, an ROI was selected surrounding part of the MT that extended 1 μm from the kinetochore. Statistical analyses were done in GraphPad Prism (methods and P values and are indicated in figure legends).

**Chromosome spreads.** Cells were synchronized as described for the bi-orientation assay. Forty five minutes after the release from monastrol into 10 μM MG132, nocodazole was added to a final concentration of 0.83 μM and incubated for 15 min. Cells were swollen by gradually increasing the concentration of Hank's Balanced Salt Solution (HBSS, Invitrogen) for 35 min. in 5% CO₂ at 37 °C and cells were fixed by gradually increasing the concentration of MeOH/Acetic acid (3:1 ratio) at room temperature. Chromosomes were visualized using DAPI. When chromosome spreads were combined with IF for Aurora B, cells were released from a thymidine block into medium containing 0.83 μM nocodazole and they were allowed to swell in 55 mM KCl₂ for 15 min in 5% CO2 at 37 °C. The cells were subsequently centrifuged onto coverslips in a 24 wells plate at 4,400 rcf for 1 min, permeabilized using PEM/T, and fixed with 4% PFA, as described above.

**Live cell imaging.** The HeLa cell lines stably co-expressing H2B-GFP and VSV-INCENPΔCEN-mCherry or VSV-CB-INCENPΔCEN-mCherry were transfected with siRNAs for INCENP and WAPL and seeded into 8-well chamber slides (Ibidi) and live cell imaging was started 5 h after thymidine release. Live cell imaging was performed using a DeltaVision microscope equipped with a CoolSNAP HQ2 camera and a × 60 objective. All image quantifications were performed using ImageJ.

**SDS–PAGE and western blotting.** Cell lines were seeded into 6 wells plates and synchronised and released from a thymidine block into medium supplemented with 20 μM STLC. Seven hours later cells were harvested and washed with PBS once followed by lysis in Laemmli buffer. SDS–polyacrylamide gel electrophoresis (SDS–PAGE) and western blotting were performed using the standard Bio-Rad protocols. The nitrocellulose membranes were blocked in TBS/0.1% Tween 20 (TBST) containing 4% milk for 30 min and incubated with the following primary antibodies: mouse anti-INCENP (1:500, Invitrogen—39–2800), mouse anti-Aurora B (1:250, BD Transduction Labs—611,083), mouse anti-α-tubulin (1:10,000, Sigma—T5168), or rabbit anti-GFP (custom made). After washing in TBST, the membranes were incubated in HRP-conjugated secondary antibodies goat anti-mouse (1:2500, Dako—P0447) or goat anti-rabbit (1:2500, Dako—P0448) in TBST-4% milk ECL (Advansta) was used as substrate of HRP and chemiluminescence was measured using an Amersham Imager 600. Uncropped scans of western blots can be found in Supplementary Fig. 7).

**BacMam virus production and transduction.** pACEBac1-CMV encoding LacI-GFP, VSV-CEN-box-GFP (aa 1-63 of INCENP), VSV-CEN-Baronase-GFP (Baronase according to ref. 41), VSV-SGO1-GFP, VSV-CEN-Sgo1 WT/N61I-GFP and sororin WT/9A-GFP (sororin WT/9A manufactured by IDT), CENP-B-mCherry (aa 1-498 of CENP-B), VSV-INCENPΔCEN-mCherry, VSV-CB-INCENPΔCEN-mCherry and VSV-MIS12-INCENPΔCEN-mCherry, were transformed into EmBacY cells to generate recombinant bacmids. For bacmid transfections, 0.5 × 10⁶ Sf9 cells per well were plated in a 6-well plate. Cells were transfected with X-tremeGENE 9 (Roche) using 3 μg recombinant bacmid (see below) and 8 μl transfection reagent in Insect-XPRESS media lacking FBS and penicillin/streptomycin. Cells were left for 5 days before harvesting the media (containing P1 virus). For production of a high titre P2 viral stock, 500 μl of P1 virus was used to inoculate a 60 ml suspension culture of log phase Sf9 cells at a density of 1.2 × 10^6 cells per ml. Cultures were grown for 4 days followed by harvesting the P2 virus by spinning down the cells at 1000 g and collecting the supernatant. Viral stocks were stored at 4 °C in the dark. For optimal viral transduction and protein expression, the HeLa cell lines were cultured in RPMI (Sigma) medium containing supplements as described above. The cells were synchronized as described for the bi-orientation assay, and viral transduction was performed 24 h prior to fixation.

**Statistical evaluation.** Statistical analysis was performed with Prism Software (Graphpad software). Data are represented as means, together with s.e.m., of either two or three independent experiments, or as dot plots of individual cells of one representative experiment. Two-tailed Student's t-test was used to compare differences between groups of cells when immunofluorescence intensity/cell was measured. When categorical outcomes were scored a Chi-squared test was used. We performed the test on binary outcomes (complete alignment vs no complete alignment = mild + severe misalignment) for the indicated control group and experimental group, to assess whether the observed rescue of complete alignment was statistically significant.

**Data availability.** All relevant data supporting the findings of this study are available within the article and its Supplementary Information files, or from the corresponding author on request.

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

## Acknowledgements

We thank Dr G. Vader for valuable input, the Lens and Kops lab members for their feedback, Dr I. Cheeseman for generous gifts of reagents and Dr G. Kops for generation of the pMELT-T601 antibody and comments on the manuscript. This research was supported by The Netherlands Organization of Scientific Research (NWO-Vici 91812610 to S.M.A.L.; and NWO-Veni 91610036 to M.A.H.).

## Author contributions

R.C.C.H., M.A.H. and S.M.A.L. conceived and designed the study. S.M.A.L. wrote the manuscript with input from R.C.C.H., M.A.H. and M.J.M.V. R.C.C.H. and M.J.M.V. performed all experiments except the ones for Supplementary Fig. 4e. M.V. generated and characterized the KNL1-pMELT-T601 antibody in the lab of G. Kops. M.J.M.V. generated the BacMam viruses.

## Additional information

**Competing interests:** The authors declare no competing financial interests.

