## [Peer Review File · Nature Communications]

Editorial Note: This manuscript has been previously reviewed at another journal that is not operating a transparent peer review scheme. This document only contains reviewer comments and rebuttal letters for versions considered at Nature Communications. Mentions of prior referee reports have been redacted.

Reviewers' Comments:

Reviewer #1:

Remarks to the Author:

The careful description of the contribution of Aurora B activity and the CEN box of INCENP to maintaining centromere cohesion is an important contribution. The proposal that spatial positioning primarily regulates checkpoint activity via positioning of KNL1 (and its PP1 binding site) relative to centromere-enriched AURKB is compelling and will certainly be of interest to the field. Finally, the discussion raises ideas that are worth further investigation. I commend the authors for their thorough and conscientious efforts to address my original concerns and I support publication of the work in Nature Communications.

Reviewer #2

[redacted]. The paper describes several novel findings. The non-catalytic role of the CPC in strengthening centromeric cohesion is particularly interesting. A requirement for the inner centromere targeting of the CPC in spindle checkpoint silencing is also novel. The authors have done a truly commendable job to address my concerns raised in the previous round of review. I found their responses more than satisfactory, and thus support the publication of this excellent study in Nature Communications.

Reviewer #3

The manuscript has elegantly demonstrated the importance of the CPC at the inner centromere in HeLa cells in supporting sister chromatid cohesion at the centromere while permitting checkpoint silencing. The conclusion will have a major influence on our understanding of how the CPC coordinates kinetochore microtubule attachment, the spindle checkpoint and sister chromatid cohesion, and greatly help clarify a long-standing enigma of why the CPC is localized at the inner centromere. My concerns and questions to the original submission were properly responded. Therefore, I support publication of this manuscript to Nature Communications [redacted] after minor revisions listed below.

Minor points

1. Ref 12. Also cite PMID: 24344188
2. Page 11, line 4. "Inner centromere confinement of Aurora B is necessary to silence the mitotic checkpoint".

I feel that this is still an overstatement of the results, as I have previously pointed out in the major point #1. To demonstrate the necessity of Aurora B at the inner centromere, the authors need to show that cytoplasmic Aurora B cannot silence the checkpoint, but I highly doubt that this is the case. The authors' data implicate the reason why the CPC must be localized to the inner centromere during mitosis, not to the kinetochore (i.e., to support the centromeric cohesion, while permitting the checkpoint silencing), but the current statement must be reworded.

Point-by-point reply to reviewers

Reviewer #1 (Remarks to the Author):

The careful description of the contribution of Aurora B activity and the CEN box of INCENP to maintaining centromere cohesion is an important contribution. The proposal that spatial positioning primarily regulates checkpoint activity via positioning of KNL1 (and its PP1 binding site) relative to centromere-enriched AURKB is compelling and will certainly be of interest to the field. Finally, the discussion raises ideas that are worth further investigation. I commend the authors for their thorough and conscientious efforts to address my original concerns and I support publication of the work in Nature Communications.

Response: We thank the reviewer for his/her positive remarks. No further action or revisions required.

Reviewer #2 (Remarks to the Author):

[redacted]. The paper describes several novel findings. The non-catalytic role of the CPC in strengthening centromeric cohesion is particularly interesting. A requirement for the inner centromere targeting of the CPC in spindle checkpoint silencing is also novel. The authors have done a truly commendable job to address my concerns raised in the previous round of review. I found their responses more than satisfactory, and thus support the publication of this excellent study in Nature Communications.

Response: We thank the reviewer for his/her positive remarks. No further action or revisions required.

Reviewer #3 (Remarks to the Author):

The manuscript has elegantly demonstrated the importance of the CPC at the inner centromere in HeLa cells in supporting sister chromatid cohesion at the centromere while permitting checkpoint silencing. The conclusion will have a major influence on our understanding of how the CPC coordinates kinetochore microtubule attachment, the spindle checkpoint and sister chromatid cohesion, and greatly help clarify a long-standing enigma of why the CPC is localized at the inner centromere. My concerns and questions to the original submission were properly responded. Therefore, I support publication of this manuscript to Nature Communications [redacted] after minor revisions listed below.

Minor points

1. Ref 12. Also cite PMID: 24344188
2. Page 11, line 4. "Inner centromere confinement of Aurora B is necessary to

silence the mitotic checkpoint”.

I feel that this is still an overstatement of the results, as I have previously pointed out in the major point #1. To demonstrate the necessity of Aurora B at the inner centromere, the authors need to show that cytoplasmic Aurora B cannot silence the checkpoint, but I highly doubt that this is the case. The authors’ data implicate the reason why the CPC must be localized to the inner centromere during mitosis, not to the kinetochore (i.e., to support the centromeric cohesion, while permitting the checkpoint silencing), but the current statement must be reworded.

Response: We understand the point raised by the reviewer and we have changed this sentence/subheading into:

“Inner centromere localization of Aurora B supports mitotic checkpoint silencing” (page 11), in accordance with the main title of the paper.

With respect to point 1: We have included this reference (ref 13 in revised manuscript.)